# Workplace stress and associated factors among vehicle repair workers in Hawassa City, Southern Ethiopia

**Hailemichael Mulugeta**[1]*, **Aiggan Tamene**[2], **Tesfaye Ashenafi**[3], **Steven M. Thygerson**[4], **Nathaniel D. Baxter**[4]

**1** Department of Public Health, College of Health Science, Debre Berhan University, Debre Berhan, Ethiopia, **2** Department of Public Health, College of Medicine and Health Sciences, Wachemo University, Shewa, Ethiopia, **3** Department of Environmental Health, College of Medical and Health Science, Hawassa University, Awasa, Ethiopia, **4** Department of Public Health, College of Life Sciences, Brigham Young University, Provo, Utah, United States of America

* hailumary464@gmail.com

## Abstract

### Introduction

Workplace stress is a public health problem worldwide. Studies focusing on work-related stress among vehicle repair workers are scarce in African countries. The current study aimed to determine the prevalence of self-reported workplace stress and associated factors among vehicle repair workers in Hawassa City, South Ethiopia.

### Methods and findings

A cross-sectional study design was employed among 347 vehicle repair workers from January 25 to February 22, 2019. Questionnaires were administered using interviews. Additional tools were used for weight and height measurements. The main findings were analyzed using descriptive statistics, bivariable, and multivariable logistic regression. The strength of association of variables was presented by odds ratio along with its 95% CI. The statistical assessments were considered significant at $p<0.05$. A total of 344 workers participated in the study. The prevalence of workplace stress among participants was 41.6% with 95% CI: (36.3–47.1). Factors associated with workplace stress were more than 10 years of work experience [AOR: 2.40; 95% CI (1.29–4.50)], work-related musculoskeletal disorder [AOR: 3.39; 95% CI (1.99–5.78)], squatting and lying work posture [AOR: 4.63; 95% CI (1.61–13.3)] and servicing large vehicles [AOR: 1.96; 95% CI (1.14–3.38)].

### Conclusion and recommendations

This study showed that the overall prevalence of work-related stress was substantially high. The independently associated factors were workers' service years, symptoms of body pain, and the work environment. Preventive measures need to be implemented in vehicle repair workshops by focusing on work environment improvements.

**Data Availability Statement:** All relevant data are within the manuscript and its Supporting Information files.

**Funding:** This research received no external funding.

**Competing interests:** The authors have declared that no competing interests exist.

## Introduction

Work-related stress is defined as the harmful emotional and physical responses that employees experience when their abilities and skills do not meet current job requirements and that challenge their ability to cope [1, 2].

Different reports indicate that workplace stress affects workers' physical health, mental health, and behavior. Stress is a predicting factor for occupational injury [3–5], work-related musculoskeletal disorder [6, 7], change of brain structure (reduction of gray matter) [8], depression [9], circulating inflammatory leukocytes [10], and high blood pressure [11].

Existing studies on the prevalence of workplace stress in Ethiopia have shown rates of 37.8% and 48.6% among healthcare professionals [12, 13], 58.2% among high school teachers [14], 60.4% among university teachers [15], and 40.4% among shoe manufacturing workers [16].

The availability of studies among vehicle repair workers to identify factors associated with workplace stress remains a significant challenge. However, findings from other groups showed that gender [17], age [15, 17], marital status [13, 18], education [12, 14, 17, 18], working experience [12, 14, 16, 19], job demands [14, 15], job security [18], poor interpersonal relationships [14, 16], smoking cigarettes [15], work hours [16, 17], job position [17], income [16, 17], alcohol use [19], and poor physical environment [16] were factors significantly associated with workplace stress.

There is a scarcity of studies on workplace stress among vehicle repair workers. A study from India revealed a workplace stress prevalence of 47% [20]. Studies among vehicle repair workshops in different countries revealed that workers suffered from occupational injuries [3], excess exposure to chemicals [20–23], work-related musculoskeletal disorder [24], occupational contact dermatitis [25, 26], and neurotoxicity [23]. Vehicle repair workers are exposed to workplace stressors including work in noisy and hot environments, poor psycho-social conditions, and awkward postures [11, 20, 27]. Such conditions may exert a negative influence on workers' health and safety.

Most worker related studies in Ethiopia do not address vehicle repair workers' stress status [28]. The majority of the vehicle repair workers did not implement effective preventive or control measures for workplace exposure [11]. Workplace stress not only affects the worker, but it has adverse effects on family income and company performance as well. Hence, evidence for planning intervention activities on stress has been limited in the country [28]. Therefore, this study aimed to estimate the prevalence of workplace stress and the associated factors among workers in vehicle repair workshops.

## Materials and methods

The current study was completed at the same time as our 2020 published study on work-related musculoskeletal disorders. Both this and our other published work studied the same study population [6]. The data collection period, study area and data collection procedures are also similar in both studies. Therefore, the published study's socio-demographic characteristics and a few other variables were similar with the current study. However, the objectives, outcomes variables and conclusions were different for each study.

### Study design, setting and period

This cross-sectional study was conducted among workers of vehicle repair workshops from January 25 to February 22, 2019, in Hawassa city. The city is the capital of the Southern

Nations, Nationalities, and Peoples' region, Ethiopia. The city is located 367 km from Addis Ababa, the capital city of Ethiopia. A total of 38 private and 3 governmental vehicle repair workshops were registered in Hawassa City in 2019 [6, 29]. Regarding service delivery, privately owned workshops provide services to vehicles owned by the government and private vehicles, but government-owned shops are established only as repair workshops for government vehicles. As of 2019, there was a total of 368 workers in 41 vehicle repair workshops in the city [6].

## The study population

The study included any worker with a minimum of 12 months of service in the industry. Exclusion criteria were: workers with any diagnosis of neurologic symptoms, congenital insensitivities (e.g. scoliosis) before starting vehicle repair work, and workers with gout, trauma, or joint disease 12 months before the data collection. A total of 347 vehicle repair workers were eligible participants. Due to inadequate work duration and disease diagnosis, 21 workers were excluded from the study.

## Study variables

Workers' stress status was the dependent variable. The independent variables include sociodemographic factors (sex, age, marital status, education, employment status, job position, service years, and monthly income); personal characteristics (physical activity, type of activity engage in after leaving work, cigarette smoking, alcohol drinking, khat chewing, job satisfaction, body mass index (BMI), work-related musculoskeletal disorders (WMSD)); and work environment characteristics (professional training, health and safety training, working hours per week, type of vehicle workshop, type of vehicle serviced, job category/responsibility, staff adequacy for work required, lifting heavy loads, work area, and work posture).

## Data collection tool and procedure

The questionnaire was administered via interviews and anthropometric measurements were used to collect study variables. The tools (S1) used to collect study variables were developed from several published papers [12, 16, 18, 22, 30–34].

Stress was assessed with the questionnaire of the Marlin Company and the American Institute of Stress Scale [30]. The tool was validated by the American Institute of Stress and used in a wide range of occupations including shoe manufacturing employees [16], oncologists [35], medical educators [36], social workers [37], and female domestic workers [38]. The stress score was calculated by analysis of the five-point Likert scale ranging from never to very often. Job satisfaction was weighed with a generic job satisfaction scale [31]. The tool was validated by previous research and has shown to be valid and useful in a wide range of occupational groups [4, 33, 34, 39–41]. The questionnaire consisted of ten items. Each item had a five-point Likert response from strongly agree to strongly disagree. The Nordic Musculoskeletal Questionnaire (NMQ-E) was used to measure WMSD symptoms in nine body regions [42]. NMQ-E is repeatable, sensitive and useful as a screening and surveillance tool and has been used in several epidemiological studies [32, 43–45]. BMI was calculated in kilograms per square meter ($kg/m^2$) [46, 47].

The study team consisted of four health and safety professional data collectors and one experienced field supervisor. The data collection tools were pretested. The interview was conducted in an isolated place to avoid disturbance and to maintain participant privacy. The study

participants' weights were measured with a standardized electronic scale with the subjects standing and wearing light clothes. The subjects' heights were measured with a portable stadiometer. The data collection activities were conducted with close follow-up by the principal investigator and supervisor.

## Data quality control

Data quality was ensured before the actual data collection. Health-related questions were translated from English to Amharic (local language) by a health professional familiar with the terminology. Vehicle repair and maintenance facilities questions were translated by an auto mechanical engineer. The translations approach was focused on cross-cultural and conceptual translations rather than the literal or linguistic equivalence of the terminologies. Then, reverse translation from the local language to English was done by an English language expert to check the consistency. Finally, discrepancies in item contents, words and meanings were checked through comparing and analyzing the translated and English language questionnaires.

The supervisor and data collectors were trained on the tools and data collection approaches. The practical session was also employed including visiting workshops, exercise data collection and rehearsing the activities. The questionnaire was pretested outside of the study area in similar workshops at Shashamane town. Modified data collection approaches (interview instead of self-administered), interview location (private location instead of at the workplace) and question order modifications were done after the pretest.

## Ethics approval and consent to participate

We conducted the study after having ethical clearance from Hawassa University College of Medicine and Health Sciences Institutional Review Board. A letter of permission was obtained from concerned governmental offices. Before performing data collection, verbal and written consent were obtained from each study participant.

## Data analysis

Error checking and coding were done before the data were entered into Epi Info version 7 software. The data was cleaned and analyzed with the statistical package for the social sciences software (SPSS) version 20 (S2). Descriptive findings were presented in frequency tables and the prevalence was reported in proportion with a 95% confidence interval (CI). The presence of a statistical relationship between different independent variables and the outcome variable was assessed using logistic regression and chi-square tests with crude odds ratio. To reduce an excessive number of independent variables and an unstable estimate in the final model, all variables with a $p$-value below 0.2 with the dependent variable in the bivariate analysis were selected for multivariable logistic regression [48–51].

A multi co-linearity assumption was checked. The result revealed a variance inflation factor <2 and tolerances >0.6. The required assumptions of the logistic regression were checked with Hosmer–Lemeshow goodness of fit test, which showed chi-squared test, $X^2 =$ 8.17, with a degree of freedom of 8 and a significance equal to 0.42. Finally, variables with a $p$-value of less than 0.05 were considered as statistically significant with 95% CI in the multivariate analysis.

## Operational definitions

**Vehicle.**   Self-propelled machinery, including large-sized (bus and truck) and other small-sized (sedan/saloon cars, minibus/minivan, and SUVs/4WDs) that do not operate on rails (such as trains or trams) which are used for the transportation of people or cargo [52].

**Vehicle repair worker.**   Male or female workers who are directly engaged in services that keep vehicle features and systems running smoothly.

**Job stress.**   A score measured using the workplace stress scale as YES (16 to 40) and NO (lower than or equal 15) [31, 36, 53].

**Work-related musculoskeletal disorders.**   A self-reported pain, ache, or discomfort for at least 2–3 workdays during the last 12 months affecting any part of the neck, shoulder, upper back, lower back, hip/thigh, knee/leg, and ankle/foot and wrist/hand. These symptoms appear at work and often disappear during rest and may continue after work ends [54]. Based on the NMQ-E screening tool, at least one symptom on any part of nine body regions over 12 months prior to the study was considered as WMSD YES; otherwise NO [32].

**Job satisfaction.**   A score measured using the job satisfaction scale as YES (32–45) and NO (10–31) [31, 33, 34].

**Body mass index.**   Body mass index is calculated as weight in kilograms divided by the square of the height in meters (kg/m$^2$) [55].

Underweight = BMI <18.50

Normal range = BMI between 18.50–24.99

Overweight = BMI between 25.00–29.99

Obese = BMI > 30.00

**Khat chewing: Khat (pronounced "cot").**   A stimulant drug derived from a shrub (*Catha edulis*) that is native to East Africa and southern Arabia. Leaves of the khat shrub are typically chewed and held in the cheek, like chewing tobacco, to release their stimulant chemicals [56].

**Nonuser.**   A person who has never used khat within 30 days preceding the study [57, 58].

**Current user.**   A person who was chewing khat within 30 days preceding the study [57, 58].

**Cigarette smoking.**   The practice of smoking at least one cigarette per day [59].

**Alcohol drinking.**   The consumption of any kind of alcohol at least two times per week [60].

**Physical exercise.**   Includes exercise in any kind of sports or physical activity at least two times per week with a duration of 30 minutes [27].

## Result

### Socio-demographic characteristics

This study included 273 participants from 38 private vehicle repair workshops and 71 participants from 3 government vehicle repair workshops. The response rate was 99.1%. Among the participants, 340 (98.8%) were male, 155 (45.1%) were 18–29 years old and the mean age was 32.3 years (standard deviation ±8.8). A majority (70.1%) were chief/senior job positions. About one-third of workers were educated up to the tertiary level and had more than 10 years of work experience in a vehicle repair workshop. The mean income of participants was 3690.7 Ethiopian birr (standard deviation ±1799.5) (Table 1).

**Table 1. Socio-demographic characteristics of vehicle repair workers in Hawassa city, Ethiopia, 2019.**

| Variables categories | Frequency (n = 344) | Percentage |
|---|---|---|
| **Marital status** | | |
| Single | 153 | 44.5 |
| Married, cohabitating | 154 | 44.8 |
| Divorce | 25 | 7.3 |
| Widowed | 12 | 3.5 |
| **Age category** | | |
| 18–29 | 155 | 45.1 |
| >29 | 189 | 54.9 |
| **Education Status** | | |
| No Formal Education | 11 | 3.2 |
| Primary Education | 104 | 30.2 |
| Secondary Education | 114 | 33.1 |
| Tertiary and above | 115 | 33.4 |
| **Years of Service** | | |
| ≤5 Years | 112 | 32.6 |
| 6–10 Years | 110 | 32.0 |
| >10 Years | 122 | 35.5 |
| **Employment status** | | |
| Permanent worker | 322 | 93.6 |
| Temporary/Contract worker | 22 | 6.4 |
| **Job Position** | | |
| Chief/Senior | 241 | 70.1 |
| Assistance | 103 | 29.9 |
| **Monthly net income Ethiopian birr** | | |
| <2500 | 95 | 27.6 |
| 2500–5000 | 196 | 57.0 |
| >5000 | 53 | 15.4 |

## Personal characteristics

Of 344 participants, 147 (42.7%) practiced regular physical exercise. The type of activities engaged in after leaving work were: the same types of work, 116 (33.7%); watching movies and reading, 142 (41.3%); and other (play games, sport, social activities), 86 (25.0%). Among the participants, the following behaviors were reported: cigarette smoking, 117 (34%); alcohol drinking, 195 (56.7%); and khat chewing, 176 (51.2%).

A total of 188 (54.7%) respondents informed their work satisfaction status as satisfied. For BMI results, 260 (75.6%) were normal, 50 (14.5%) were overweight, 24 (7.0%) were under-weight, and 10 (2.9%) were obese. Regarding WMSD, 164 (47.7%) reported at least one the symptoms in the last 12 months.

## Work environment characteristics

The majority of participants, 228 (66.3%) had no training in general workplace safety. Among the participants, 150 (43.6%) were mechanical repair workers and 132 (38.4%) spent more than 48 hours in their job per week. Other workplace considerations reported were insufficient staff, 161 (46.8%); working in an area of inadequate space, 193 (56.1%); and routinely lifting loads greater than 20 kg, 147 (42.7%) (Table 2).

**Table 2. Work environment characteristics of vehicle repair workers in Hawassa City, Ethiopia, 2019.**

| Variables categories | Frequency (n = 344) | Percent (%) |
|---|---|---|
| **Professional Training** | | |
| Yes | 156 | 45.3 |
| No | 188 | 54.7 |
| **Health and Safety-related training** | | |
| Yes | 116 | 33.7 |
| No | 228 | 66.3 |
| **Type of Vehicle serviced** | | |
| Large sized (bus or truck) | 115 | 33.4 |
| Small sized (sedan/saloon cars, minibus/minivan, and SUVs/4WDs) | 229 | 66.6 |
| **Job category/ responsibility** | | |
| Mechanical repair | 150 | 43.6 |
| Electrical repair | 71 | 20.6 |
| Body beating/wielding | 77 | 22.4 |
| Spray painting | 32 | 9.3 |
| Other # | 14 | 4.1 |
| **Inadequate number of staff** | | |
| Yes | 161 | 46.8 |
| No | 183 | 53.2 |
| **Type of floor at work site** | | |
| Concrete | 130 | 37.8 |
| Gravel | 125 | 36.3 |
| No ground cover | 89 | 25.9 |
| **Inadequate workspace** | | |
| Yes | 193 | 56.1 |
| No | 151 | 43.9 |
| **Heavy loads job more than 20kg routinely** | | |
| Yes | 147 | 42.7 |
| No | 197 | 57.3 |
| **Most commonly adopted work posture** | | |
| Standing | 128 | 37.2 |
| Bending | 103 | 29.9 |
| Kneeling | 49 | 14.2 |
| Sitting | 15 | 4.4 |
| Squatting, prone position on the ground | 49 | 14.2 |

*#Note*: *Repair associated activities (like seat repair, radiator repair work, etc.. . .)*

## Prevalence and factors associated with work-related stress

The overall prevalence of work-related stress among vehicle repair workers in Hawassa city, Ethiopia was found to be 41.6% [95% CI (36.3–47.1)].

In the bivariate logistic regression analysis, all variables were analyzed to select the candidate independent variables for multivariable logistic regression. Variables with a *p*-value of less than 0.2 in the bivariate logistic regression analysis were selected. Socio-demographic variables (educational status, employment status and service year), personal related characteristics (physical exercise, type of activity engage in after leaving the work, alcohol drinking, Khat chewing and work-related musculoskeletal disorder) and work environment (health and

safety-related training, type of vehicle serviced, type of vehicle workshop, inadequate number of staff, inadequate workspace, a heavy loads job more than 20kg routinely and work posture) were fitted and selected for the multivariable logistic regression model. Service year [AOR: 2.40; 95% CI (1.29–4.50)], work-related musculoskeletal disorder [AOR: 3.39; 95% CI (1.99–5.78)], servicing of large-sized vehicles [AOR: 1.96; 95% CI (1.14–3.38)] and working in squatting and lying work posture [AOR: 4.63; 95% CI (1.61–13.3)] were found to be associated with a higher likelihood of work-related stress (Table 3).

## Discussion

Workplace stress is a public health problem for employees worldwide. Studies focusing on work-related stress in African countries are scarce, especially among vehicle repair workers [28, 61]. This study revealed the prevalence and factors associated with work-related stress.

The present study indicated that 41.6% [95% CI (36.3–47.1)] of the participants experienced work-related stress. The current prevalence was in line with the study reported among vehicle repair workers in India (47%) [20] and shoe manufacturing workers in Ethiopia (40.4%) [16]. However, our findings had a higher prevalence than among textile factory workers in Republic of Congo (28%) [19] and the secondary school teacher study in Malaysia (32.3%) [62]. It was also lower than those of studies among female domestic workers in Lucknow, India (77%) [38] social workers in the United States of America (81.8%) [37] and medical educators in Pakistan (94%) [36]. Due to the scarcity of similar studies, prevalence estimates are difficult to compare, especially with findings from a similar work environment using similar measures. That said, the discrepancy of the findings might be due to work environment differences (i.e., for all studies except vehicle repair workers in India), local context including perceptions and traditions, measuring tools (Congo and Malaysia), and individual living standards, which could have effects on findings being either high or low prevalence [63].

In the current study, years of service, work-related musculoskeletal disorders, servicing large-sized vehicles and work postures were associated with work-related stress. Participants with >10 years of service [AOR: 2.40; 95% CI (1.29–4.50)], had twice the odds of having workplace stress compared with those who worked ≤5 years. This finding is inconsistent with other studies among workers in health care, academic, and manufacturing industries where those who had more work experience reported a lower odds of workplace stress [12, 14, 16, 19]. A possible explanation for the difference could be the exposure of vehicle repair workers to chemicals like petroleum products and heavy metals for long periods [20–23]. This might increase the likelihood of stress development by affecting changes in biochemical levels [11, 21] and nerve damage or peripheral neuropathy [20]. Peripheral neuropathy is the inflammation or damage of the peripheral nerves, which connect the central nervous system to the muscles, glands, senses, and other internal organs, causing numbness in the hands and feet [64]. This type of nerve damage might contribute to stress development [65].

Moreover, this study showed work-related musculoskeletal disorders were significantly associated with work-related stress. Workers who reported musculoskeletal disorders [AOR: 3.39; 95% CI (1.99–5.78)] had more than three times the odds of having workplace stress than individuals who did not report MSDs. One reason may be that many stressed individuals might report musculoskeletal disorders and justify that stress and WMSDs have a relationship [66, 67].

Also, this finding indicated that servicing large-sized vehicles [AOR: 1.96; 95% CI (1.14–3.38)] and squatting and lying work postures [AOR: 4.63; 95% CI (1.61–13.3)] were associated with workplace stress. The report showed that individuals who serviced buses and trucks had more odds of having workplace stress than workers servicing small-sized vehicles. Workers

**Table 3. Multivariate logistic regression of factors associated with work-related stress among vehicle repair workers in Hawassa city, Ethiopia, May 2019.**

| Variable name | Stress status (n = 344) | | Crude OR (95%CI) | Adjusted OR (95%CI) |
|---|---|---|---|---|
| | Yes (%) | No (%) | | |
| **Education Status** | | | | |
| No Formal | 5 (45.5) | 6 (54.5) | 080 (0.23–2.79) | 1.22 (0.29–5.21) |
| Primary | 53 (51.0) | 51 (49.0) | 1.48 (0.43–5.16) | 0.69 (0.37–1.29) |
| Secondary | 41 (36.0) | 73 (64.0) | 1.35 (0.39–4.67) | 1.13 (0.60–2.11) |
| Tertiary | 44 (38.3) | 71 (61.7) | 1 | 1 |
| **Employment status** | | | | |
| Permanent | 137 (42.5) | 185 (57.5) | 1 | 1 |
| Temporary/Contract | 6 (27.3) | 16 (72.7) | 2.0 (0.75–5.18) | 2.05 (0.68–6.18) |
| **Year of service** | | | | |
| ≤5 | 54 (48.2) | 58 (51.8) | 1 | 1 |
| 6–10 | 44 (40.0) | 66 (60.0) | 1.40 (0.82–2.38) | 2.09 (1.10–3.98) * |
| >10 | 45 (36.9) | 77 (63.1) | 1.59 (0.95–2.69) | 2.40 (1.29–4.50)* |
| **Alcohol drinking** | | | | |
| Yes | 88 (45.1) | 107 (54.9) | 1.41(0.91–2.17) | 1.22 (0.72–2.08) |
| No | 55 (36.9) | 94 (63.1) | 1 | 1 |
| **Khat chewing** | | | | |
| Non-user | 62 (36.9) | 106 (63.1) | 1 | 1 |
| Current user | 81 (46.0) | 95 (54.0) | 1.46 (0.95–2.24) | 1.48 (0.87–2.52) |
| **Activity engaged after leaving work** | | | | |
| The same types of work | 60 (51.7) | 56 (48.3) | 1.8 (1.10–2.96) * | 1.23 (0.68–2.20) |
| Watching movies and Reading | 53 (37.3) | 89 (62.7) | 2.0 (1.13–3.55) * | 1.26 (0.63–2.5) |
| Play game, sport, social activities | 30 (34.9) | 56 (65.1) | 1 | 1 |
| **Regular physical exercise** | | | | |
| Yes | 55 (37.4) | 92 (62.6) | 1 | 1 |
| No | 88 (44.7) | 109 (55.3) | 0.74 (0.48–1.45) | 0.87 (0.52–1.47) |
| **WMSD** | | | | |
| Yes | 93 (56.7) | 71 (43.3) | 3.41 (2.17–5.34) * | 3.39 (1.99–5.78) * |
| No | 50 (27.8) | 130 (72.2) | 1 | 1 |
| **Safety-related training** | | | | |
| Yes | 39 (33.6) | 77 (66.4) | 1 | 1 |
| No | 104(45.6) | 124 (54.4) | 1.65 (1.04–2.64) * | 1.28 (0.75–2.18) |
| **Type of vehicle workshop** | | | | |
| Private owned | 120 (44.0) | 153(56.0) | 1.63 (0.94–2.84) | 1.06 (0.48–2.34) |
| Government-owned | 23 (32.4) | 48 (67.6) | 1 | 1 |
| **Type of vehicle serviced** | | | | |
| Large sized | 55 (47.8) | 60 (52.2) | 1.47 (0.93–2.31) | 1.96 (1.14–3.38) * |
| Small sized | 88 (38.4) | 141 (61.6) | 1 | 1 |
| **Inadequate workspace** | | | | |
| Yes | 90 (46.6) | 103 (53.4) | 1.62 (1.04–2.50) * | 1.18 (0.69–2.01) |
| No | 53 (35.1) | 98 (64.9) | 1 | 1 |
| **Routinely lifting greater than 20 kg** | | | | |
| Yes | 70 (47.6) | 77 (52.4) | 1.54 (1.00–2.38) * | 1.22 (0.73–2.04) |
| No | 73 (37.1) | 124 (62.9) | 1 | 1 |
| **Inadequate number of staff** | | | | |
| Yes | 77 (47.8) | 84 (52.2) | 1.63 (1.06–2.50) * | 1.30 (0.77–2.19) |
| No | 66 (36.1) | 117 (63.9) | 1 | 1 |

(*Continued*)

**Table 3.** (Continued)

| Variable name | Stress status (n = 344) | | Crude OR (95%CI) | Adjusted OR (95%CI) |
|---|---|---|---|---|
| | Yes (%) | No (%) | | |
| **Most commonly adopted work posture** | | | | |
| Standing | 65 (50.8) | 63 49.2) | 1 | 1 |
| Sitting | 2 (13.3) | 13 (86.7) | 6.7 (1.45–30.9) * | 5.35 (1.07–26.6) * |
| Kneeling | 26 (53.1) | 23 (46.9) | 0.91 (0.47–1.76) | 0.82 (0.39–1.75) |
| Bending | 38 (36.9) | 65 (63.1) | 1.76 (1.03–3.00) * | 2.05 (1.06–3.96) * |
| Squatting, prone position on the ground | 12 (24.5) | 37 (75.5) | 3.18 (1.52–6.65) * | 4.63 (1.61–13.3) * |

*p-value <0.05*

servicing large-sized vehicles might be well experienced and have long service years compared to small-sized vehicle repair workers. Exposure duration to chemicals might have a contribution to the stress difference among workers group [20–23]. The odds of workplace stress among employees who reported working their job in a squatting and lying posture was more than four times than those who did not engage in such posture. The possible explanation could be that poor physical environment and ergonomics hazards contribute to workplace stress [16, 66].

Though this study was able to provide important data on work-related stress among vehicle repair workers, several limitations are noted. Since this finding was reported on existing data and focused on workplace stress as the outcome, psychosocial work factors, especially those related to the organizational risk factors, were missed. We believe that the study provided a reasonably accurate assessment of work-related stress and associated factors.

## Conclusions

This study shows that the overall prevalence of work-related stress was substantially high among vehicle repair workers in Hawassa City, Ethiopia. The most important independent risk factors identified by a multivariable logistic regression model were work experience, work-related musculoskeletal disorder, servicing large-sized vehicles, and squatting and lying work postures. In other words, these factors were workers' years of service, symptoms of pain on various body regions, and the work environment. Therefore, preventive measures need to be implemented on the reduction of workplace stress among vehicle repair workers by focusing on work environment improvements. Further studies in the broadest context are recommended.

## Supporting information

**S1 File. English version of consent form and questionnaire.**
(DOCX)

**S1 Data. Data set used for analysis.**
(SAV)

## Acknowledgments

We appreciate the Hawassa University, College of Medicine and Health Sciences, Research and Community Service Office that allowed us to conduct the study. We thank the vehicle

repair workers, managers and data collectors for giving their valuable time and cooperation during the data collection period.

## Author Contributions

**Conceptualization:** Hailemichael Mulugeta.

**Data curation:** Hailemichael Mulugeta, Aiggan Tamene, Tesfaye Ashenafi.

**Formal analysis:** Tesfaye Ashenafi, Steven M. Thygerson, Nathaniel D. Baxter.

**Investigation:** Hailemichael Mulugeta, Aiggan Tamene.

**Methodology:** Hailemichael Mulugeta.

**Project administration:** Hailemichael Mulugeta.

**Software:** Hailemichael Mulugeta.

**Supervision:** Hailemichael Mulugeta.

**Validation:** Hailemichael Mulugeta.

**Visualization:** Hailemichael Mulugeta, Steven M. Thygerson, Nathaniel D. Baxter.

**Writing – original draft:** Hailemichael Mulugeta, Aiggan Tamene, Tesfaye Ashenafi, Steven M. Thygerson, Nathaniel D. Baxter.

**Writing – review & editing:** Hailemichael Mulugeta.

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
