## [Decision Letter · Decision Letter 0]

18 Jan 2021

PONE-D-20-37255

Workplace stress and Associated Factors among Vehicle Repair Workers in Hawassa City, Southern Ethiopia

PLOS ONE

Dear Dr. Mulugeta,

Thank you for submitting your manuscript to PLOS ONE. After careful consideration, we feel that it has merit but does not fully meet PLOS ONE’s publication criteria as it currently stands. Therefore, we invite you to submit a revised version of the manuscript that addresses the points raised during the review process.

An expert in the field handled your manuscript, and we are appreciative of their time and contributions. Although interest was found in your study, several concerns arose that require your attention. Notably, there are serious concerns about the methodological approach to validation of your experimental parameters. Please address ALL of the reviewer's comments in your revised manuscript. 

We look forward to receiving your revised manuscript.

Kind regards,

Frank T. Spradley

Academic Editor

PLOS ONE

2) Please include additional information regarding the survey or questionnaire used in the study and ensure that you have provided sufficient details that others could replicate the analyses. For instance, if you developed a questionnaire as part of this study and it is not under a copyright more restrictive than CC-BY, please include a copy, in both the original language and English, as Supporting Information, or include a citation if it has been published previously.

3) You have mentioned that the questionnaire was pre-tested, but not whether it was validated. Please clarify if it was validated. If this did not occur, please provide the rationale for not validating the questionnaire.

4) In your discussions and conclusions please take care to avoid statements implying causality from correlational research. For example, avoid the use of terms such as "risk factors" or “effects” or “resulted in." Instead consistently use terms such as "associated with" or "associations."

5)  Thank you for submitting the above manuscript to PLOS ONE. During our internal evaluation of the manuscript, we found significant text overlap between your submission and the following previously published works, some of which you are an author.

- https://www.hindawi.com/journals/jeph/2020/9472357/

- https://pubmed.ncbi.nlm.nih.gov/32165812/

Please revise the manuscript to rephrase the duplicated text, cite your sources, and provide details as to how the current manuscript advances on previous work. Please note that further consideration is dependent on the submission of a manuscript that addresses these concerns about the overlap in text with published work.

Reviewers' comments:

Reviewer's Responses to Questions

**Comments to the Author**

1. Is the manuscript technically sound, and do the data support the conclusions?

Reviewer #1: Partly

2. Has the statistical analysis been performed appropriately and rigorously? 

Reviewer #1: No

3. Have the authors made all data underlying the findings in their manuscript fully available?

Reviewer #1: No

4. Is the manuscript presented in an intelligible fashion and written in standard English?

Reviewer #1: No

5. Review Comments to the Author

Reviewer #1: The cross-sectional study aimed to examine the association of demographic, personal, and work characteristics with work stress among Ethiopian vehicle repair workers. Although this topic has been widely studied in developed countries, work condition in this population is rarely described and the study topic is important for work condition improving in developing countries. Nevertheless, this study has several major limitations that should be addressed. Frist, the authors included a lot of variables but in the method section, the statistic model was not clearly described. Second, this study would provide substantial contribution to occupational health literature if the validated and widely used work stress model can be incorporated (e.g., job control-demand model and effort-reward imbalance model). Unfortunately the risk factors for work stress in this study were not measured by validated tools. Below I listed my comments, and I hope they will help improve the manuscript.

Abstract:

The tools used to measure workplace stress and categories of independent variables should be clearly stated.

2.3 study variables:

BMI and WMSD were listed as independent variables, while musculoskeletal symptoms were listed in line 117-118 with other outcome variables. It is unclear which variables were treated as dependent and which were independent variables.

3.3 Job stress and 3.5 job satisfaction:

Please briefly describe how the cutoff point was chosen or validated in previous literature.

2.2 study population:

Among the 368 vehicle workers, 344 participated in the study. Were the 24 workers excluded from the study due to inadequate work duration, or diagnosis of diseases, or disagree with study participation? Please clarify.

3.4 Work-related musculoskeletal disorders (WMSD) is measured by self-report symptoms. It is not clear what is the difference between WMSD here and the symptoms measured by NMQ-E in line 116-117, while the NMQ-E score seems not being included in the regression models.

Table 4 and 4.4 Prevalence and factors associated with work-related stress:

Please indicate what variables were included in the crude and adjusted models. Does the crude model the same as bivariate model mentioned in 2.4 data analysis line 148-151? In addition, the outcome variable in the logistic model in table 4 was work-related stress. It is unclear whether job satisfaction and musculoskeletal symptoms were outcome variables or independent variables? Because they were explained in the 2.3 study variable section, but were not seen in the logistic model.

5. Discussion:

The authors compared the prevalence of work stress in line 237-246. However, it is unclear whether work stress was measured by the same tool or questionnaire in these cited studies. If not, were the prevalence rates in these studies comparable?

In the last section of discussion (line 270-274), another limitation is that work psychosocial factors were not assessed using validated tools in the literature, such as job demand-control model, and effort-reward imbalance model. This study chose work stress as the outcome, but missed psychosocial work factors as risk factors.

6. conclusions

The authors are recommended to provide more objective implications for how to improve work environment of vehicle repairing workers. For example, for work postures, how should vehicle repairing company improve their settings to avoid awkward work positions?

6. PLOS authors have the option to publish the peer review history of their article (what does this mean?). If published, this will include your full peer review and any attached files.

Reviewer #1: No

---

## [Author Response · Author response to Decision Letter 0]

24 Feb 2021

We would like to thank the reviewers for the time and effort put on to revise our manuscript `` Prevalence of self-reported workplace stress and associated factors among vehicle repair workers in Hawassa City, South Ethiopia`` in detail. We believe that the comments have identified important areas that required improvement. We have revised in detail as PLOS One format. In addition, we did revision based on the editor`s comments and incorporated new information than what you commented. After completion of the suggested edits, the revised manuscript has benefitted from an improvement in the overall presentation and clarity.

Dear Reviewers thank you for your comment and support through all processes. We included new variables that we obtained after recategorized variables from the data set (SPSS). The following points are the response to your idea.

Issues from reviewers: 

1. Abstract 

• Abstract:

The tools used to measure workplace stress and categories of independent variables should be clearly stated.

Response

Accepted the comment: Thank you for the critical observation.

I have included the following information.

Questionnaires were administered using interviews. Additional tools were used for weight and height measurements.

Issues from reviewers: 

2.Method section

2.3 study variables:

BMI and WMSD were listed as independent variables, while musculoskeletal symptoms were listed in line 117-118 with other outcome variables. It is unclear which variables were treated as dependent and which were independent variables

3.3 Job stress and 3.5 job satisfaction: Please briefly describe how the cutoff point was chosen or validated in previous literature.

Response

Dear Reviewer: It is right and I accepted the comment. I have described the issues in the manuscript document.

Both body mass index (BMI) and work-related musculoskeletal disorders (WMSD) were independent variables. Weight in kilogram and height in centimeter were measured. BMI was calculated in kilogram per meter square (kg/m2). A digital weight scale was used for assessing weight, and height.

WMSD symptoms were measured in nine body regions using the extended version of the Nordic Musculoskeletal Questionnaire (NMQ-E). It is a valid, repeatable, sensitive and useful as a screening and surveillance tool.

# Both Job stress and job satisfaction were briefly described under the data collection tool and procedure topic.

Job stress:

We used the American Institute of Stress Scale to measure stress. It is a questionnaire; it was designed to apply a workplace stress data collection tool to assess job stress levels of the American workforce. The tool was validated by the American Institute of Stress. It is used to assess job stress among workers in a wide range of occupations and has been used in several studies for stress level evaluation. The questionnaire consists of eight items describing how often a respondent feels toward his or her job. The scale is in the five-point Likert response format, ranging from never (scored 1) to very often (scored 5). High scores are indicative of higher levels of job stress except item numbers 6, 7, and 8 are reverse-scored. The total scores are interpreted as follows: scores of 15 and below: relatively calm and stress isn’t much of an issue, 16–20: fairly low, 21–25: moderate levels of work stress, 26–30: severe levels of work stress, and 31–40: potentially dangerous level of work stress.

Interpretation of the current study: Job Stress was YES if the score measured was 16-40 and NO if the result was lower than or equal to 15. 

 Job satisfaction:

We used the Generic job satisfaction scale to assess the level of satisfaction. It is a questionnaire; was developed to collect job satisfaction levels among peoples in Ontario, Canada. The tool was validated by different studies and it is also used in a wide range of occupational groups. The questionnaire consists of ten items. Each item had a 5-point Likert response from strongly agree to strongly disagree. The scores are interpreted 42-50: very high, 39-41: high, 32-38: average, 27-3l: low and 10-25: very low

Interpretation of the current study: Using the job satisfaction scale YES if a measured score was 32–45 and NO if the result was lower than or equal to 10–31. No result greater than 45 and less than 10 scores was found in our study.

Issues from reviewers: 

2.2 study population:

Among the 368 vehicle workers, 344 participated in the study. Were the 24 workers excluded from the study due to inadequate work duration, or diagnosis of diseases, or disagree with study participation? Please clarify.

Response

Dear Reviewer, thank you very much. I accepted the comment and wrote the reason under the study population and result topic.

A total of 21 workers didn’t fit the eligibility criteria and three participants disagreed to participate. A total of 344 vehicle repair workers participated in this study giving a response rate of 99.1%.

Issues from reviewers: 

3.4 Work-related musculoskeletal disorders (WMSD) is measured by self-report symptoms. It is not clear what is the difference between WMSD here and the symptoms measured by NMQ-E in line 116-117, while the NMQ-E score seems not being included in the regression models.

Response

Dear Reviewer! I accepted the comment and revised it.

WMSDs are defined as injuries or disorders of the muscles, nerves, tendons, joints, cartilage, and spinal discs that may be associated with the work environment and performance of work. And due to the disorder, workers might report pain, ache, or discomfort in nine body regions. These symptoms appear at work and often disappear during rest or exit in life long time. 

We did not apply any diagnostic instrument without the screening tool of NMQ-E. All response options of the questionnaire were dichotomous (yes/no). At least one symptom on any body part of nine body region over 12 months prior to the study was considered as WMSD Yes Otherwise NO. 

WMSD is described under the operational definition topic based and WMSD Yes/No response was included in the regression models and presented in Table 3.

Issues from reviewers: 

3. Result

Table 4 and 4.4 Prevalence and factors associated with work-related stress: Please indicate what variables were included in the crude and adjusted models. Does the crude model the same as the bivariate model mentioned in 2.4 data analysis line 148-151? In addition, the outcome variable in the logistic model in table 4 was work-related stress. It is unclear whether job satisfaction and musculoskeletal symptoms were outcome variables or independent variables? Because they were explained in the 2.3 study variable section but were not seen in the logistic model.

Response

Accepted the comment: Topic 2.4 data analysis and 4.4 Prevalence and factors associated with work-related stress needed clarity. We revised it to make it more understandable. The following points might answer your question. 

Method: 

The bivariate logistic regression model shows the relation between independent variables with workplace stress (outcome or dependent variable). Therefore; one independent variable was interred into bivariate logistic regression at a time; then the model gave the relationship between one independent variable and a given outcome variable. But it shows a crude. 

Multivariable logistic regression model shows the relation between an independent variable with workplace stress through adjusting cofounder /other independent variables. Therefore; more than two independent variables were entered into the model; then the model gave the relationship between one independent and a given outcome variable through adjusting cofounder or other explanatory independent variables at a time in a different row of a table. But the output of the model is affected or unstable if a large number of variables were entered into the model at a time. Since there were many independent variables in the current study, each variable was analyzed using bivariate logistic regression to select candidates for multivariable logistic regression. This step was help us to reduce an excessive number of variables and an unstable estimate in the final model. Finally, all independent variables with a p-value below 0.2 with the dependent variable in the bivariate analysis were selected for multivariable analysis.

Result: 

Socio-demographic variables (educational status, employment status and service year), personal related characteristics (physical exercise, type of activity engage in after leaving the work, alcohol drinking, chewed Khat and work-related musculoskeletal disorder) and work environment (health and safety-related training, type of vehicle serviced, type of vehicle workshop, inadequate number of staff, inadequate workspace, a heavy loads job more than 20kg routinely and work posture) were fitted and selected for the multivariable logistic regression model. Service year [AOR: 2.40; 95% CI (1.29-4.50)], work-related musculoskeletal disorder [AOR: 3.39; 95% CI (1.99-5.78)], serviced large-sized vehicle [AOR: 2.03; 95% CI (1.16-3.54)] and working in squatting and lying work posture [AOR: 4.63; 95% CI (1.61-13.3)] were found to be associated with a higher likelihood of work-related stress (Table 3). 

Both job satisfaction and Musculoskeletal symptoms were explained as Work-related musculoskeletal disorder (WMSD) were independent variables. Job satisfaction was not selected for Multivariable logistic regression because of p- value>0.2 in bivariate logistic regression. According to the operational definition, a single musculoskeletal symptom count as WMSD yes. Therefore; musculoskeletal symptoms were found in the logistic regression in named of WMSD (table 3)

Issues from reviewers: 

4. Discussion:

The authors compared the prevalence of work stress in line 237-246. However, it is unclear whether work stress was measured by the same tool or questionnaire in these cited studies. If not, were the prevalence rates in these studies comparable?

In the last section of discussion (line 270-274), another limitation is that work psychosocial factors were not assessed using validated tools in the literature, such as job demand-control model, and effort-reward imbalance model. This study chose work stress as the outcome but missed psychosocial work factors as risk factors.

Response

Accepted the comment: I have tried to find a similar study and used a similar tool. The studies, what we found based on similar tools were used to comparing. But it was difficult to find both similar study and the same tool. So that; we included other stress studies but the inconsistency was explained in the discussion part.

---

## [Decision Letter · Decision Letter 1]

12 Mar 2021

PONE-D-20-37255R1

Workplace Stress and Associated Factors among Vehicle Repair Workers in Hawassa City, Southern Ethiopia

PLOS ONE

Dear Dr. Mulugeta,

Thank you for submitting your manuscript to PLOS ONE. After careful consideration, we feel that it has merit but does not fully meet PLOS ONE’s publication criteria as it currently stands. Therefore, we invite you to submit a revised version of the manuscript that addresses the points raised during the review process.

We look forward to receiving your revised manuscript.

Kind regards,

Frank T. Spradley

Academic Editor

PLOS ONE

Journal Requirements:

Reviewers' comments:

Reviewer's Responses to Questions

**Comments to the Author**

1. If the authors have adequately addressed your comments raised in a previous round of review and you feel that this manuscript is now acceptable for publication, you may indicate that here to bypass the “Comments to the Author” section, enter your conflict of interest statement in the “Confidential to Editor” section, and submit your "Accept" recommendation.

Reviewer #1: (No Response)

2. Is the manuscript technically sound, and do the data support the conclusions?

Reviewer #1: Yes

3. Has the statistical analysis been performed appropriately and rigorously? 

Reviewer #1: Yes

4. Have the authors made all data underlying the findings in their manuscript fully available?

Reviewer #1: Yes

5. Is the manuscript presented in an intelligible fashion and written in standard English?

Reviewer #1: No

6. Review Comments to the Author

Reviewer #1: I appreciated the authors' effort to revise the manuscipt, and I feel that my comments have been addressed and the clarity of the manuscript has improved. I have only two more commnets:

1. There is an obvious error in the revision page 6 section Study variables: ”workers’ stress status was the dependent variable. The dependent variable include…”

I think the authors mean “The independent variables include…”

2. This manuscript needs a thorough editing. For example, the limitation section has been duplicated. There are also many grammar errors that need to be corrected. Since I am not an English native speaker, I suggest tha authors to look for professional English-editing services.

7. PLOS authors have the option to publish the peer review history of their article (what does this mean?). If published, this will include your full peer review and any attached files.

Reviewer #1: No

---

## [Author Response · Author response to Decision Letter 1]

20 Mar 2021

Point by point response to the reviewers

We would like to thank the reviewers for the time and effort put on to revise our manuscript `` Prevalence of self-reported workplace stress and associated factors among vehicle repair workers in Hawassa City, South Ethiopia`` in detail. We accepted your two issue and did revision. In addition, English language professional fixed the language related faults in the document.

---

## [Editor Report · Decision Letter 2]

23 Mar 2021

Workplace Stress and Associated Factors among Vehicle Repair Workers in Hawassa City, Southern Ethiopia

PONE-D-20-37255R2

Dear Dr. Mulugeta,

We’re pleased to inform you that your manuscript has been judged scientifically suitable for publication and will be formally accepted for publication once it meets all outstanding technical requirements.

Kind regards,

Frank T. Spradley

Academic Editor

PLOS ONE

---

## [Editor Report · Acceptance letter]

25 Mar 2021

PONE-D-20-37255R2 

Workplace Stress and Associated Factors among Vehicle Repair Workers in Hawassa City, Southern Ethiopia 

Dear Dr. Mulugeta:

I'm pleased to inform you that your manuscript has been deemed suitable for publication in PLOS ONE. Congratulations! Your manuscript is now with our production department. 

Kind regards, 

on behalf of

Dr. Frank T. Spradley 

Academic Editor

PLOS ONE